# Wave Propagation in Rotating Functionally Graded Microbeams Reinforced by Graphene Nanoplatelets

**DOI:** 10.3390/molecules26175150

**Published:** 2021-08-25

**Authors:** Tianyu Zhao, Yu Ma, Jiannan Zhou, Yanming Fu

**Affiliations:** 1School of Science, Northeastern University, Shenyang 110819, China; zhaotianyu@mail.neu.edu.cn (T.Z.); 2000299@stu.neu.edu.cn (Y.M.); 2Technology Center of Shenyang Customs, Shenyang 110016, China; tank261014@163.com; 3Laboratory Management Center, Shenyang Sport University, Shenyang 110102, China

**Keywords:** graphene nanoplatelets, wave propagation, rotation, microbeams, functionally graded

## Abstract

This paper presents a study on wave propagation in rotating functionally graded (FG) microbeams reinforced by graphene nanoplatelets (GPLs). The graphene nanoplatelets (GPLs) are considered to distribute in the diameter direction of the micro-beam in a gradient pattern, which leads to the functionally graded structure. By using the Halpin-Tsai micromechanics model and the rule of mixture, the effective material properties of the microbeam are determined. According to the Euler-Bernoulli beam theory and nonlocal elasticity theory, the rotating microbeams are modeled. A comprehensive parametric study is conducted to examine the effects of rotating speed, GPL distribution pattern, GPL length-to-thickness ratio, GPL length-to-width ratio, and nonlocal scale on the wavenumber, phase speed and group speed of the microbeam. The research findings can play an important role on the design of rotating graphene nanoplatelet (GPL) reinforced microbeams for better structural performance.

## 1. Introduction

In recent years, microbeams are strongly applied in micro components and products, such as microrotors, micro air vehicles and so on. Generally, microbeams used in microproducts are minute-sized, and their nonlocal effect needs to be considered. Different structural performance is shown compared to that of macrostructures [1,2,3]. Thus, many scholars paid attention to the analysis of microbeams.

Based on a quasi-3D beam theory, Tlidji et al. [4] investigated the free vibration of a functionally graded (FG) microbeam. Considering a thermal and magnetic environment, Esen et al. [5] studied the vibration characteristics of cracked microbeams. Zhang et al. [6] conducted vibration analysis of a porous microbeam subject to a moving harmonic load. According to the non-Fourier heat conduction model, Abouelregal et al. [7] carried out the thermos elastic performance of spinning microbeams. Dynamic responses of microbeams with a shallow arched shape and clamp boundary in pressure sensors are presented by Najar et al. [8]. Chen et al. [9] investigated the buckling and dynamic stability of FG microbeams embedded in elastic medium. On the basis of a modified couple stress theory, Fang et al. [10] studied the size-dependent vibration behaviors of a spinning microbeam. Ghommem et al. [11] presented a study on nonlinear dynamic characteristics of rotary nano crystalline material microbeams under electric excitation. Shenas et al. [12] examined the thermal effect on the post-buckling behavior of a spinning pre-twisted microbeam. Yao et al. [13] investigated the transverse free vibration and wave propagation of Timoshenko microbeams with an axial motion.

To sum up, most research are focused on the vibration performance of microbeams. It is known that the microstructures are always fail due to the vibration. As an important method of nondestructive inspection, the evaluation of wave propagation characteristics in microbeams is quite necessary to be carried out.

To enhance the structural performance, composite materials are always adopted in a real project. As one of the most promising materials, graphene nanoplatelet reinforced material has attracted the attention of many scholars. Yang and Zhao [14,15,16,17,18,19,20,21] studied the vibration characteristics of some rotating structures in engineering. Zhou et al. [22] investigated the flutter behaviors of a porous cylindrical panels reinforced by GPLs under the supersonic flow. Wang et al. [23] studied the nonlinear vibrations of a GPL reinforced metal foam circular cylindrical shell. Kitipornchai et al. [24] presented a study on the elastic buckling of an FG beam with different pore distributions. Based on the first order shear deformation theory, Reddy et al. [25] investigated the vibrations of GPL reinforced thick multi-layer, moderately thick and thin plates. Sahmani et al. [26] examined the nonlocal vibration behavior of a GPL reinforced microplate with pores. Employing a numerical-based generalized differential quadrature method, Lori et al. [27] analysis the free vibrations of microdisks reinforced with GPLs. By using the semi-analytical formulation, Li et al. [28] presented wave propagation in annular plates reinforced by GPLs. Amirabadi et al. [29] studied wave propagation analytically in spinning thick cylindrical shells. Gao et al. [30] investigated wave propagation in a GPL reinforced FG porous plate.

Form the above analysis, it can be understood that wave propagation in FG microbeams reinforced by GPLs needs to be studied. This paper aims to investigate the wave propagation characteristics of rotating GPL reinforced microbeams. The parametric analysis is conducted in detail. The results can be an aid in the design of microbeams.

## 2. Theoretical Modeling

### 2.1. Physical Model

The rotating microbeam model can be expressed as a cantilever beam with a displacement perpendicular to the rotating plane. The rotating GPLs reinforced microbeam system can be described by the cylindrical coordinate system *O*-*xθz*. The radius (*R_B_*) and length (*L_B_*) of the microbeam are depicted in the figure as shown in Figure 1, where the origin is located at the center of the microbeam, the *z* axis is consistent with the rotation direction, and the *x* axis is consistent with the extension direction of the axis.

### 2.2. Material Parameters

The rotating microbeam studied in this paper is composed of polymer as matrix and GPLs as reinforcement. According to Halpin-Tsai model [31,32], the effective Young’s modulus of the microbeam (*E_B_*) is
(1)$$E_{B}\left(r\right)=\frac{3}{8}\frac{1+\xi_{L}\left(\frac{E_{GPL}/E_{M}-1}{E_{GPL}/E_{M}+\xi_{L}}\right)V_{GPL}}{1-\left(\frac{E_{GPL}/E_{M}-1}{E_{GPL}/E_{M}+\xi_{L}}\right)V_{GPL}}E_{M}+\frac{5}{8}\frac{1+\xi_{W}\left(\frac{E_{GPL}/E_{M}-1}{E_{GPL}/E_{M}+\xi_{W}}\right)V_{GPL}}{1-\left(\frac{E_{GPL}/E_{M}-1}{E_{GPL}/E_{M}+\xi_{W}}\right)V_{GPL}}E_{M}$$
where *E_GPL_* and *E_M_* are the Young’s modulus of GPLs and the polymer matrix, respectively.

For specifying the geometrical characteristics of GPLs in the microbeam, (*ξ_L_*, *ξ_W_*) is given by
(2)$$E_{M}\left\{\begin{array}{l} \xi_{L}=2L_{G}/t_{G}\\ \xi_{W}=2W_{G}/t_{G} \end{array}\right.$$
in which *L_G_*, *W_G_*, and *t_G_* are the length, width, and thickness of GPLs in the microbeam.

The GPL volume fractions of the microbeam (*V_GPL_*) is
(3)$$V_{GPL}=\frac{W_{GPL}}{W_{GPL}+\rho_{GPL}\left(1-W_{GPL}\right)/\rho_{M}}$$
where *ρ_GPL_* and *ρ_M_* are the mass density of GPLs and the polymer matrix. *W_GPL_* is the weight fraction of GPLs.

The GPL volume fractions of the the microbeam (*V_GPL_*) are considered to vary along the radius direction on basis of the specific GPL distribution patterns, determined by
(4)$$V_{GPL}\left(r\right)=\left\{\begin{array}{ll} \frac{r^{2}}{R^{2}}V_{1} & \mathrm{Pattern}\;X\\ V_{2} & \mathrm{Pattern}\;U\\ \left(1-\frac{r^{2}}{R^{2}}\right)V_{3} & \mathrm{Pattern}\;O \end{array}\right.$$
where *V*_1_, *V*_2_ and *V*_3_ are the GPL volume fraction indices which can be obtained by giving the total weight fraction of GPLs (*W_TGPL_*) in the microbeam.

As shown in Figure 2, different GPL distribution patterns are considered. Figure 2a expresses the positive parabolic distribution of GPL in the microbeam, where the weight fraction of GPL is zero at the center of the circle and is the largest at the surface; Figure 2b indicates the uniform distribution of GPL in the microbeam, where the weight fraction of GPL remains constant along the radius; Figure 2c shows the negative parabola distribution of GPL in the microbeam, where the weight fraction of GPL is zero at the surface and the maximum at the center.

According to the mixing rule, the mass density (*ρ_B_*) and Poisson’s ratio along (*ρ_B_*) the radius of the microbeam are obtained as
(5)$$\begin{array}{l} \rho_{B}\left(r\right)=V_{GPL}\rho_{GPL}+V_{M}\rho_{M}\\ \mu_{B}\left(r\right)=V_{GPL}\mu_{GPL}+V_{M}\mu_{M} \end{array}$$
where *μ_GPL_* and *μ_M_* are Poisson’s ratio of GPLs and the polymer matrix. *V_M_* is the volume fractions of polymer matrix in the microbeam, which is given as
(6)$$V_{M}+V_{GPL}=1$$

### 2.3. Governing Equations

The rotating microbeam can be expressed as a cantilever beam perpendicular to the rotating plane. Based on the Euler-Bernoulli beam theory, the axial and transverse displacement fields of the rotating microbeam can be expressed as Equations (7) and (8)
(7)$$u(x,y,z,t)=u^{0}-z\frac{\partial w}{\partial x}$$
(8)$$w\left(x,y,z,t\right)=w\left(x,t\right)$$
where *u*_0_ and *w* are the axial and transverse displacements of point (*x*, 0) on the plane (*z* = 0) in the microbeam. According to the von Kármán strain, the only nonzero strain is
(9)$$\varepsilon_{xx}=\frac{\partial u}{\partial x}=\frac{\partial u^{0}}{\partial x}-z\frac{\partial^{2}w}{\partial x^{2}}$$

Due to Euler–Bernoulli beam theory, the differential equations of motion are Equations (10) and (11)
(10)$$\frac{\partial^{2}M}{\partial x^{2}}+\frac{\partial }{\partial x}\left(T(x)\frac{\partial w}{\partial x}\right)=H\frac{\partial^{2}w}{\partial t^{2}}$$
(11)$$\frac{\partial^{2}M}{\partial x^{2}}+\frac{\partial }{\partial x}\left(T(x)\frac{\partial w}{\partial x}\right)=H\frac{\partial^{2}w}{\partial t^{2}}$$
where *H* is an integral constant defined, *Q* is the axial force, *M* is the bending moment and *T*(*x*) is the axial force of the rotating microbeam due to centrifugal stiffening, which is given as
(12)$$H=\int_{A}\rho_{B}(r)dA=2\pi \int_{0}^{R_{B}}r\rho_{B}(r)dr$$
(13)$$T(x)=\int_{x}^{L}H\Omega^{2}xdx=\pi \Omega^{2}\left(L^{2}-x^{2}\right)\int_{0}^{R_{B}}r\rho_{B}(r)dr$$
(14)$$Q=\int_{A}\sigma_{xx}dA$$
(15)$$M=\int_{A}z\sigma_{xx}dA$$
where the stress *σ_xx_* in Equations (14) and (15) is given by nonlocal theory, which is given as
(16)$$\sigma_{xx}-{\left(e_{0}a\right)}^{2}\frac{\partial^{2}\sigma_{xx}}{\partial x^{2}}=E_{B}(r)\varepsilon_{xx}$$

Taking the integral form of Equations (14) and (15) for Equation (16), and substituting Equations (14) and (15), we can obtain
(17)$$Q-{\left(e_{0}a\right)}^{2}\frac{\partial^{2}Q}{\partial x^{2}}=A_{0}\frac{\partial u}{\partial x}$$
(18)$$M-{\left(e_{0}a\right)}^{2}\frac{\partial^{2}M}{\partial x^{2}}=-B\frac{\partial^{2}w}{\partial x^{2}}$$
where *A*_0_ and *B*_0_ are defined integral constants, and they are given as:(19)$$A_{0}=\int_{A}E_{B}(r)dA=2\pi \int_{0}^{R_{B}}rE_{B}(r)dr$$
(20)$$B=\int_{A}E_{B}(r)z^{2}dA=\pi \int_{0}^{R_{B}}r^{3}E_{B}(r)dr$$

With the help of the nonlocal constitutive relations and motion equations proposed above, the torque can be expressed by a given generalized displacement. We get Equation (21) by substituting Equation (18) into Equation (11).
(21)$$M=-B\frac{\partial^{2}w}{\partial x^{2}}+{\left(e_{0}a\right)}^{2}\left[H\frac{\partial^{2}w}{\partial t^{2}}-\frac{\partial }{\partial x}\left(T(x)\frac{\partial w}{\partial x}\right)\right]$$

Substituting in Equation (21) into Equation (11), the following equation of motion of the non-locally rotating microbeam can be obtained as
(22)$$-B\frac{\partial^{4}w}{\partial x^{4}}+{\left(e_{0}a\right)}^{2}\frac{\partial }{\partial x^{2}}\left[H\frac{\partial^{2}w}{\partial t^{2}}-\frac{\partial }{\partial x}\left(T(x)\frac{\partial w}{\partial x}\right)\right]+\frac{\partial }{\partial x}\left(T(x)\frac{\partial w}{\partial x}\right)=H\frac{\partial^{2}w}{\partial t^{2}}$$

In the rotating microbeam, we replace *T*(*x*) with *T_max_* (*x* = 0) where the maximum force is. Then we can get
(23)$$T_{\max}=\pi \Omega^{2}L^{2}\int_{0}^{R_{B}}r\rho_{B}(r)dr$$

Substituting Equation (23) into Equation (22), the governing equation of the lateral displacement (*w*(*x*, *t*)) of the rotating cantilever beam can be expressed as a constant coefficient nonlocal partial differential equation.
(24)$$B\frac{\partial^{4}w}{\partial x^{4}}-T_{\max}\frac{\partial^{2}w}{\partial x^{2}}+T_{\max}{\left(e_{0}a\right)}^{2}\frac{\partial^{4}w}{\partial x^{4}}-H{\left(e_{0}a\right)}^{2}\frac{\partial^{2}w}{\partial t^{2}\partial x^{2}}+H\frac{\partial^{2}w}{\partial t^{2}}=0$$

## 3. Solution Procedure

In order to analyze the dispersion characteristics of the wave in the microbeam, we assume that the harmonic wave solution of the displacement field *w*(*x*, *t*) can be expressed in complex form as
(25)$$w(x,t)=\sum_{n=1}^{N}\overset{⌢}{w}\left(x,\omega_{n}\right)e^{-j\left(kx-\omega_{n}t\right)}$$
where $\overset{⌢}{w}$ is the frequency domain amplitude of the bending deformation of microbeam, *k* is the wave number, *ω_n_* is the wave angular frequency at sampling points, *N* is the number of samples and $j=\sqrt{-1}$.

Substituting the displacement of Equation (25) into Equation (24) to eliminate the time variable, Equation (26) is given as
(26)$$\begin{array}{l} \sum_{n=1}^{N}\left[B\frac{\partial^{4}\overset{⌢}{w}\left(x,\omega_{n}\right)}{\partial x^{4}}-T_{\max}\frac{\partial^{2}\overset{⌢}{w}\left(x,\omega_{n}\right)}{\partial x^{2}}+T_{\max}{\left(e_{0}a\right)}^{2}\frac{\partial^{4}\overset{⌢}{w}\left(x,\omega_{n}\right)}{\partial x^{4}}\right.\\ \left.+H{\left(e_{0}a\right)}^{2}\omega^{2}\frac{\partial^{2}\overset{⌢}{w}\left(x,\omega_{n}\right)}{\partial x^{2}}-H\omega^{2}\overset{⌢}{w}\left(x,\omega_{n}\right)\right]e^{j\omega_{n}t}=0 \end{array}$$

Equation (26) is satisfied for each *n*. Therefore, it can be abbreviated as:(27)$$B\frac{d^{4}\overset{⌢}{w}}{dx^{4}}-T_{\max}\frac{d^{2}\overset{⌢}{w}}{dx^{2}}+T_{\max}{\left(e_{0}a\right)}^{2}\frac{d^{4}\overset{⌢}{w}}{dx^{4}}+H{\left(e_{0}a\right)}^{2}\omega^{2}\frac{d^{2}\overset{⌢}{w}}{dx^{2}}-H\omega^{2}\overset{⌢}{w}=0$$

In order to simplify the analysis, we express this equation in a dimensionless form. Now given the new variable is expressed as
(28)$$\bar{x}=\frac{x}{L_{B}}\;,\bar{w}=\frac{\overset{⌢}{w}}{L_{B}}\;,\;\frac{d\bar{w}}{d\bar{x}}=\frac{d\overset{⌢}{w}}{dx},\frac{d^{2}\bar{w}}{d{\bar{x}}^{2}}=L_{B}\frac{d\overset{⌢}{w}}{dx},\frac{d^{3}\bar{w}}{d{\bar{x}}^{3}}=L_{B}{}^{2}\frac{d\overset{⌢}{w}}{dx^{3}},\frac{d^{4}\bar{w}}{d{\bar{x}}^{4}}=L_{B}{}^{3}\frac{d^{4}\overset{⌢}{w}}{dx^{4}},\tau =\frac{e_{0}a}{L_{B}}$$

The dimensionless form of Equation (27) is as follows:(29)$$B\frac{1}{L_{B}{}^{3}}\frac{d^{4}\bar{w}}{d{\bar{x}}^{4}}-T_{\max}\frac{1}{L_{B}}\frac{d^{2}\bar{w}}{d{\bar{x}}^{2}}+T_{\max}\tau^{2}\frac{1}{L_{B}}\frac{d^{4}\bar{w}}{d{\bar{x}}^{4}}+H\omega^{2}\tau^{2}L_{B}\frac{d^{2}\bar{w}}{d{\bar{x}}^{2}}-L_{B}H\omega^{2}\bar{w}=0$$

Defining a new variable:(30)$$\omega_{B}=\frac{1}{L_{B}{}^{2}}\sqrt{\frac{B}{H}}$$

Substituting Equation (30) into Equation (29) and sorting it out, Equation (31) is given as
(31)$$\left[1+\frac{1}{2}{\left(\frac{\Omega }{\omega_{B}}\right)}^{2}\tau^{2}\right]\frac{d^{4}\bar{w}}{d{\bar{x}}^{4}}+\left[{\left(\frac{\omega }{\omega_{B}}\right)}^{2}\tau^{2}-\frac{1}{2}{\left(\frac{\Omega }{\omega_{B}}\right)}^{2}\right]\frac{d^{2}\bar{w}}{d{\bar{x}}^{2}}-{\left(\frac{\omega }{\omega_{B}}\right)}^{2}\bar{w}=0$$

Since the differential equation is a constant coefficient equation, the form of solution can be set as
(32)$$\bar{w}=\bar{W}e^{-jkx}\left(\bar{W}\neq 0\right)$$

Substituting the solution into Equation (31), the discrete characteristic Equation (33) of GPLs reinforced rotating microbeam can be obtained
(33)$$\left[1+\frac{1}{2}{\left(\frac{\Omega }{\omega_{B}}\right)}^{2}\tau^{2}\right]k^{4}+\left[{\left(\frac{\omega }{\omega_{B}}\right)}^{2}\tau^{2}-\frac{1}{2}{\left(\frac{\Omega }{\omega_{B}}\right)}^{2}\right]k^{2}-{\left(\frac{\omega }{\omega_{B}}\right)}^{2}\bar{w}=0$$
where *k* is the wave number. Solving Equation (33), it can be concluded that:(34)$$k=\pm \sqrt{\frac{-\left[\frac{1}{2}{\left(\frac{\Omega }{\omega_{B}}\right)}^{2}-\tau^{2}{\left(\frac{\omega }{\omega_{B}}\right)}^{2}\right]\pm \sqrt{{\left[\frac{1}{2}{\left(\frac{\Omega }{\omega_{B}}\right)}^{2}-\tau^{2}{\left(\frac{\omega }{\omega_{B}}\right)}^{2}\right]}^{2}+4\left[1+\tau^{2}\frac{1}{2}{\left(\frac{\Omega }{\omega_{B}}\right)}^{2}\right]{\left(\frac{\omega }{\omega_{B}}\right)}^{2}}}{2\left[1+\frac{1}{2}\tau^{2}{\left(\frac{\Omega }{\omega_{B}}\right)}^{2}\right]}}$$

The wave number *k* is mainly determined by the nonlocal scaling parameter (*e*_0_*a*), the rotation speed of the beam Ω, and the circular frequencies *ω* and *ω* are determined. Meanwhile, phase velocities *C_p_* and group velocities *C_g_* are obtained from Equation (34).
(35)$$C_{p}=real\left(\frac{\omega }{k}\right),C_{g}=real\left(\frac{\partial \omega }{\partial k}\right)$$

## 4. Numerical Results and Discussion

According to the numerical experiments, the radius *R_B_* of the microbeam is assumed to be *R_B_* = 1 × 10^−6^m and the length *L_B_* is assumed to be *L_B_* = 20*R_B_* The matrix is epoxy resin, and its young’s modulus, density and Poisson’s ratio are *E_M_* = 2.85 GPa, *ρ_M_* = 1.2 × 10^3^kg/m^3^ and *μ_M_* = 0.34, respectively. The reinforcement is GPLs and its young’s modulus, density and Poisson’s ratio are *E_GPL_* = 1.01 Tpa, *ρ_GPL_* = 1.06 × 10^3^ kg/m^3^, and *μ_GPL_* = 0.006. In the following analysis and discussion, if there is no additional explanation, the nonlocal scale *τ* = 0 nm, the total weight fraction of GPLs in the microbeam *W_TGPL_* = 1%, the length thickness ratio of GPLs *L_G_*/*t_G_* = 10^3^, the length width ratio of GPLs *L_G_*/*W_G_* = 2, and the GPLs distribution mode is U Pattern. In this case, the following variables are given:(36)$$\omega_{0}=\Omega_{0}=C_{0}=\frac{1}{L_{B}{}^{2}}\sqrt{\frac{B}{H}}$$

Among them, the current given values are used for material parameters, which will not change during the analysis of the following parts.

Figure 3a indicates that the wave number varies with dimensionless frequency *ω*/*ω_0_* at a dimensionless rotation speed of 0 for different GPLs mass fraction (*W_TGPL_* = 0.00%, 0.33%, 0.67%, and 1.00%). It can be seen that the wave number decreases with the increase of the mass fraction of GPLs in the microbeam. It can be concluded that adding more graphene to the microbeam can effectively reduce the wave number and enhance the performance of microbeam. Meanwhile, the wave number will increase with the increase of dimensionless frequency and the wave number is nonlinear with the increase of dimensionless frequency, which means that the wave number changes dispersedly. It implies that the shape of the wave will change when it propagates. Figure 3b shows the wave number when the dimensionless rotational speed Ω/Ω_0_ is 2. Compared with Figure 3a, when the rotational speed increases, the wave number tends to increase linearly with the increase of frequency, which indicates that the wave number changes tend to be nondispersive, which means that the wave will not change its shape when it propagates.

Figure 4a displays that the variation of the phase velocity and the group velocity with dimensionless frequency under the mass fraction of GPLs (*W_TGPL_* = 0.00%, 0.33%, 0.67%, and 1.00%), when the dimensionless rotation speed Ω/Ω_0_ is 0. One can see that the group velocity and phase velocity increases with the increase of the weight fraction of GPLs in the microbeam. It is implied that adding more graphene to the microbeam can effectively rise the phase velocity and the group velocity and enhance the performance of microbeam. At the same time, the group velocity and phase velocity increases with the rise of dimensionless frequency. When the frequency is small, the group velocity and phase velocity increase faster; When the frequency is large, the group velocity and phase velocity increase slowly. In addition, the group velocity and phase velocity are non-linear with the increase in frequency at different weight fractions of GPLs, which means that the wave number changes dispersedly and the shape of the wave will change when it propagates. Figure 4b shows that the variation of the phase velocity and the group velocity with dimensionless frequency when the dimensionless rotational speed Ω/Ω_0_ is 2. Compared with Figure 4a, when the dimensionless rotation speed is increased, the difference is that the group velocity and phase velocity increase slowly, if the dimensionless frequency is small; if the dimensionless frequency is large, the group velocity and phase velocity increase faster. If the rotational speed continues to increase, the difference between group velocity and phase velocity will continue to decrease, and both of them will reach saturation to constant speed. This is the characteristic of all nondecentralized systems. In extreme cases, it can be said that their frequencies become equal and constant for all waves, which means that the wave will not change its shape when it propagates.

Figure 5 shows that the wave number varies with dimensionless frequency *ω*/*ω_0_* for different length thickness ratio of GPLs (*L_G_*/*t_G_* = 10, 10^2^, 10^3^, and 10^4^). At the dimensionless rotation speeds of 0 and 2, it can be observed that the wave number will decline with the increase of the length thickness ratio of GPLs in the microbeam. When the content of GPLs is the same, higher length thickness ratio means thinner GPLs. It indicates that it is useful to adding thinner GPLs into microbeams to increase the wave number and enhance the microbeam.

The change of the phase velocity and the group velocity with dimensionless frequency for different length-thickness ratio of GPLs (*L_G_*/*t_G_* = 10, 10^2^, 10^3^, and 10^4^) at the dimensionless rotation speed of 0 and 2. can be revealed in Figure 6. It can be seen that the group velocity and phase velocity go up with the increase of the length thickness ratio of GPLs in the microbeam. It can be confirmed that adding thinner GPLs into microbeams is a good way to get better performance.

Figure 7 reflects that the wave number varies with dimensionless frequency for different GPLs length width ratio (*L_G_*/*W_G_* = 1, 2, 4, and 8) under the dimensionless rotation speeds of 0 and 2. It suggests that the wave number rise with the increase of the length width ratio of GPLs in the microbeam. If the content of GPLs is fixed, low length-width ratio ratio means that the GPLs can obtain a larger surface area. It signifies that we should take GPLs with larger surface area into microbeams to enhance the microbeam.

Figure 8 displays that the phase velocity and the group velocity varies with dimensionless frequency for different GPLs length-width ratio (*L_G_*/*W_G_* = 1, 2, 4, and 8) under the dimensionless rotation speeds of 0 and 2. It is clarified that the group velocity and phase velocity diminish with the increase of the length thickness ratio of GPLs in the microbeam. It indicates that the phase velocity and the group can be increased by taking GPLs with larger surface area into microbeams, which is beneficial to enhance the microbeam.

Figure 9 presents that the wave number changes with dimensionless frequency for different distribution patterns of GPLs (pattern X, pattern U, and pattern O) under the dimensionless rotation speeds of 0 and 2. It suggests that when the distribution pattern of GPLs is patteren O, the corresponding wave number is the largest. When the distribution pattern is patteren X, the corresponding wave number is the smallest. This is to say that, more GPLs dispersed on the surface of the microbeam can further reduce the wave number and improve the performance when the content of GPLs is fixed.

Figure 10 displays the variation of the phase velocity and the group velocity with dimensionless frequency for different distribution patterns of GPLs (pattern X, pattern U, and pattern O) when the dimensionless rotation speeds are 0 and 2. It can be confirmed that when the distribution pattern of GPLs is patteren X, the phase velocity and the group velocity are the largest. When the distribution pattern is patteren O, the phase velocity and the group velocity is the smallest. It can be thus ensured that we can disperse more GPLs disperse on the surface of the micro beam to improve the performance when the content of GPLs is fixed.

Figure 11 indicates the changes of the wave number with dimensionless frequency for different nonlocal scales (*τ* = 0.1, 0.2, 0.3, and 0.4) at the dimensionless rotation speeds of 0 and 2 are illustrated in Figure 8. Result shows that the wave number will increase with the increase of the nonlocal scale in the microbeam at the dimensionless rotation speed is 0. In addition, if the rotational speed is Ω/Ω_0_ = 2, when the frequency is *ω*/*ω*_0_ < 1, the wave number increases with the increase of the nonlocal scale; when the frequency is *ω*/*ω*_0_ > 1, the wave number decrease with the increase of the nonlocal scale. This indicates that the influence of the nonlocal scale on the wave number will change due to the increase of rotation speed, which we must attach great importance to.

Figure 12a depicts that the variation of the phase velocity and the group velocity with dimensionless frequency for different nonlocal scales (*τ* = 0.1, 0.2, 0.3, and 0.4) at the dimensionless rotation speeds of 0 and 2. As can be seen, the group velocity and phase velocity will decrease with the increase of the nonlocal scale in the microbeam at the dimensionless rotation speed of 0. However, it is different from the trend of the dimensionless rotation speed of 0 when the rotation speed is increased. When the frequency is *ω*/*ω*_0_ < 1, the phase velocity still decreases with the increase of the nonlocal scale; however, the phase velocity is tuned to decrease with the increase of the nonlocal scale when the frequency is *ω*/*ω*_0_ > 1. In addition, the group velocity increase with the increase of the nonlocal scale in the microbeam which is also different compared with before. Meanwhile, the curve fluctuates when *ω*/*ω*_0_ is in (0.2, 0.5). Thus, it is of great importance that we need to consider the dimensionless frequency, the nonlocal scale and the dimensionless rotation speed all at the same time in practice.

## 5. Conclusions

In this study, based on the nonlocal theory, the Young’s modulus and mass density of GPLs reinforced microbeams were estimated according to the improved Halpin-Tsai micromechanical model and mixing law. Meanwhile, the rotating microbeam is modeled as a Euler-Bernoulli beam. Using Euler-Bernoulli beam theory, the motion differential equation of the rotating microbeam with nonlocal scale effect is derived, which provides a powerful model for analyzing the wave dispersion characteristics of the rotating microbeam. In this paper, the wave number, group velocity and phase velocity varying with dimensionless frequency under different rotational speed, weight fraction of GPLs, length thickness ratio of GPLs, length weight ratio of GPLs and distribution patterns of GPLs, and nonlocal scaling coefficient are studied. The main conclusions are as follows:(1)Adding more GPLs into the microbeam can significantly improve the performance.(2)Thinner GPLs are suitable to enhance structure performance of the microbeam than ordinary GPLs.(3)GPLs with larger surface area have a better enhancement effect on the microbeam.(4)With more GPLs distributed on the surface of the microbeam, the reinforcement effect can be improved while the content of GPLs remains unchanged.(5)Nonlocal scaling is a very important parameter, because its influence on the wave number the phase velocity and the group will change with the dimensionless frequency and speed, which we must take it into consideration.(6)Rotational speed plays an important role in wave propagation. When rotational speed is 0, the wave number, the group velocity and phase velocity are changed nonlinearly with the frequency, which means that the wave number varies dispersedly. However, when the rotational speed increases, the wave number, group velocity, and phase velocity tend to increase linearly, which indicates that the wave will not change its shape when it propagates.

## Figures and Tables

**Figure 1 molecules-26-05150-f001:**
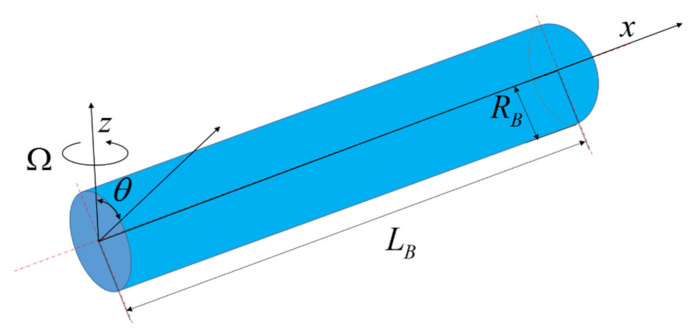
Model and coordinate system.

**Figure 2 molecules-26-05150-f002:**
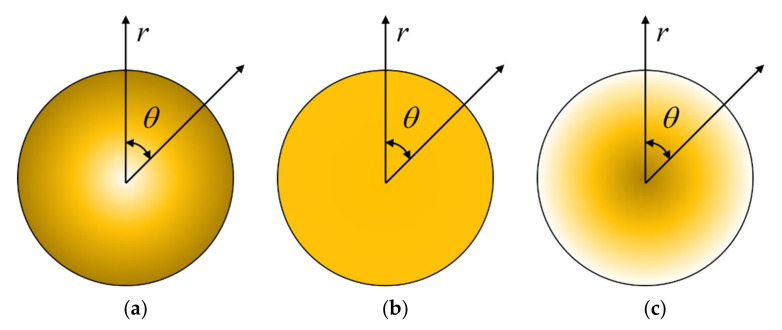
GPL distribution patterns for the microbeam. (**a**) Pattern X; (**b**) Pattern U; (**c**) Pattern O.

**Figure 3 molecules-26-05150-f003:**
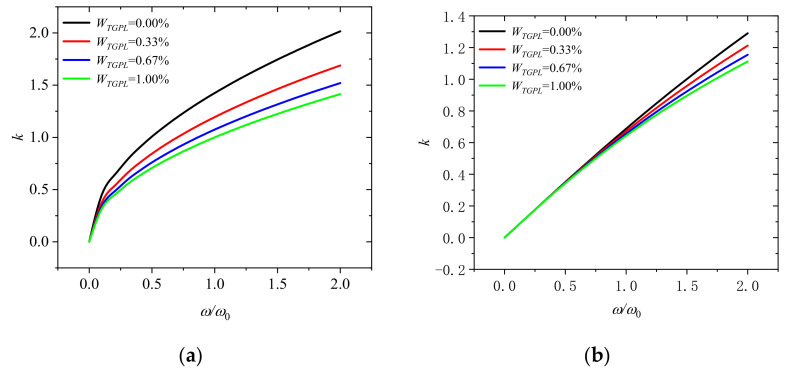
Wavenumber dispersion in rotating microbeam with different weight fraction of GPLs (*W_TGPL_* = 0.00%, 0.33%, 0.67%, and 1.00%) for (**a**) Ω/Ω_0_ = 0 and (**b**) Ω/Ω_0_ = 2.

**Figure 4 molecules-26-05150-f004:**
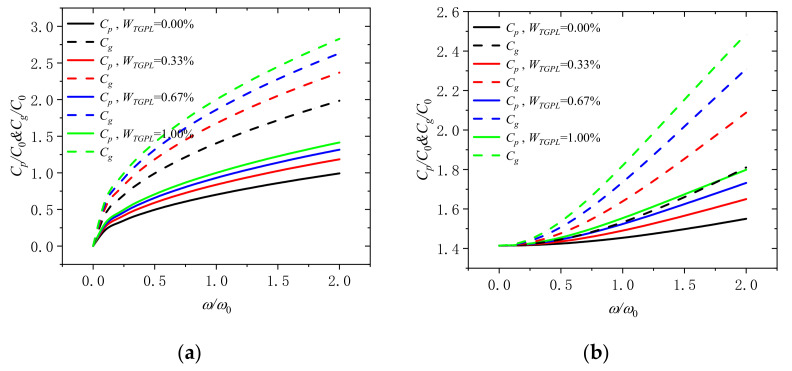
Phase speed and group speed in rotating microbeam with different weight fraction of GPLs (*W_TGPL_* = 0.00%, 0.33%, 0.67%, and 1.00%) for (**a**) Ω/Ω_0_ = 0 and (**b**) Ω/Ω_0_ = 2.

**Figure 5 molecules-26-05150-f005:**
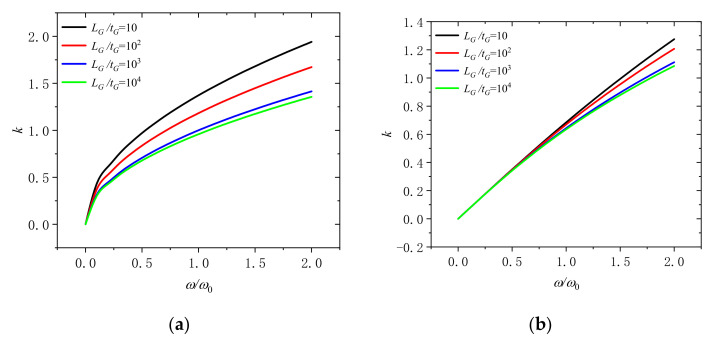
Wavenumber dispersion in a rotating microbeam with different length thickness ratio of GPLs (*L_G_*/*t_G_* = 10, 10^2^, 10^3^, and 10^4^) for (**a**) Ω/Ω_0_ = 0 and (**b**) Ω/Ω_0_ = 2.

**Figure 6 molecules-26-05150-f006:**
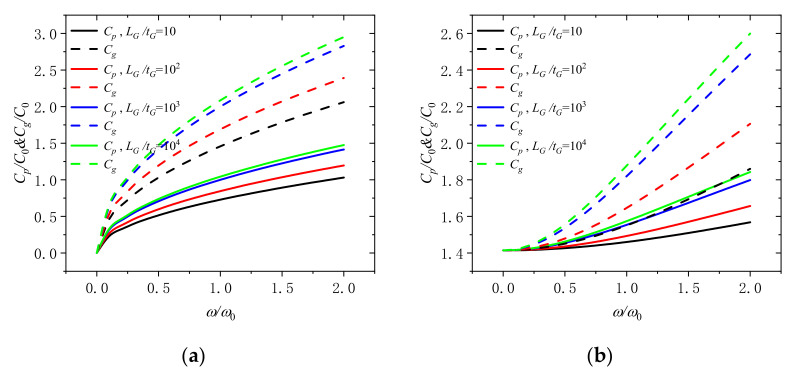
Phase speed and group speed in a rotating microbeam with different length thickness ratio of GPLs (*L_G_*/*t_G_* = 10, 10^2^, 10^3^, and 10^4^) for (**a**) Ω/Ω_0_ = 0 and (**b**) Ω/Ω_0_ = 2.

**Figure 7 molecules-26-05150-f007:**
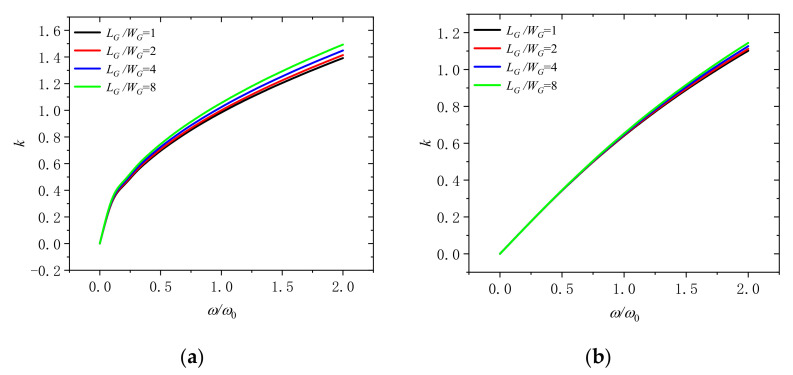
Wavenumber dispersion in a rotating microbeam with different length width ratio of GPLs (*L_G_*/*W_G_* = 1, 2, 4, and 8) for (**a**) Ω/Ω_0_ = 0 and (**b**) Ω/Ω_0_ = 2.

**Figure 8 molecules-26-05150-f008:**
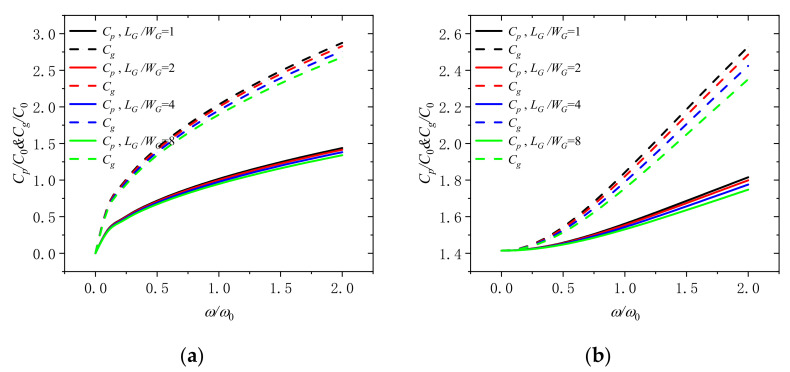
Phase speed and group speed in a rotating microbeam with different length width ratio of GPLs (*L_G_*/*W_G_* = 1, 2, 4, and 8) for (**a**) Ω/Ω_0_ = 0 and (**b**) Ω/Ω_0_ = 2.

**Figure 9 molecules-26-05150-f009:**
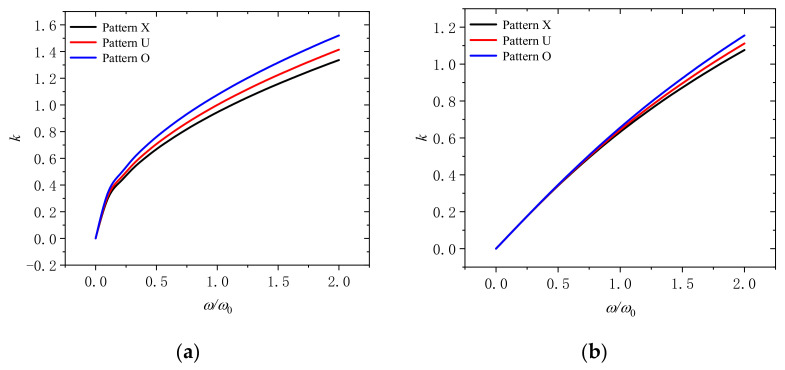
Wavenumber dispersion in a rotating microbeam with different distribution patterns of GPLs (pattern X, pattern U, and pattern O) for (**a**) Ω/Ω_0_ = 0 and (**b**) Ω/Ω_0_ = 2.

**Figure 10 molecules-26-05150-f010:**
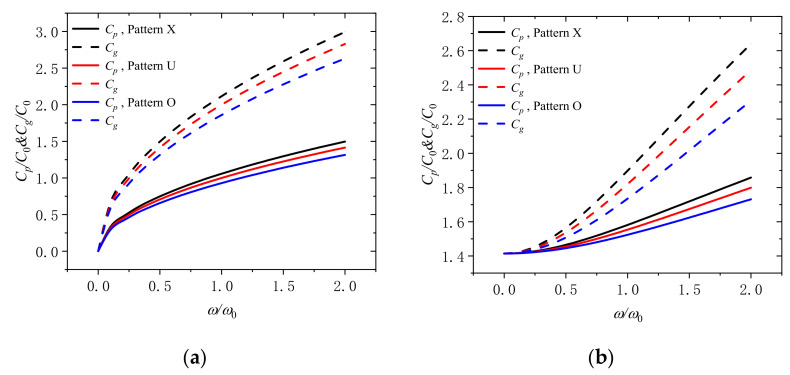
Phase speed and group speed in rotating micro-beam with different distribution patterns of GPLs (Pattern X, Pattern U, Pattern O) for (**a**) Ω/Ω_0_ = 0 and (**b**) Ω/Ω_0_ = 2.

**Figure 11 molecules-26-05150-f011:**
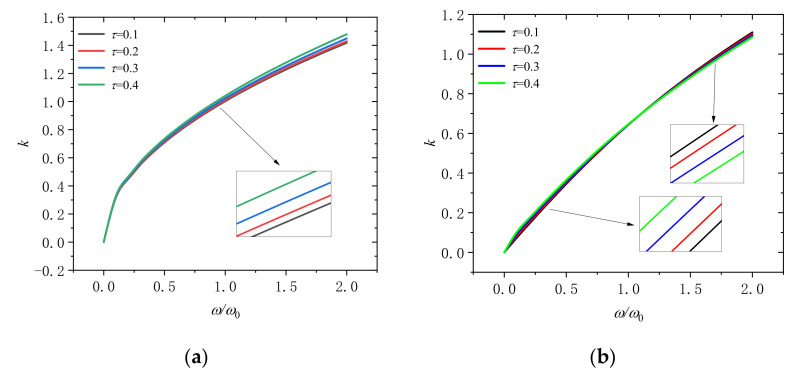
Wavenumber dispersion in a rotating microbeam with different nonlocal scales (*τ* = 0.1, 0.2, 0.3, and 0.4) for (**a**) Ω/Ω_0_ = 0 and (**b**) Ω/Ω_0_ = 2.

**Figure 12 molecules-26-05150-f012:**
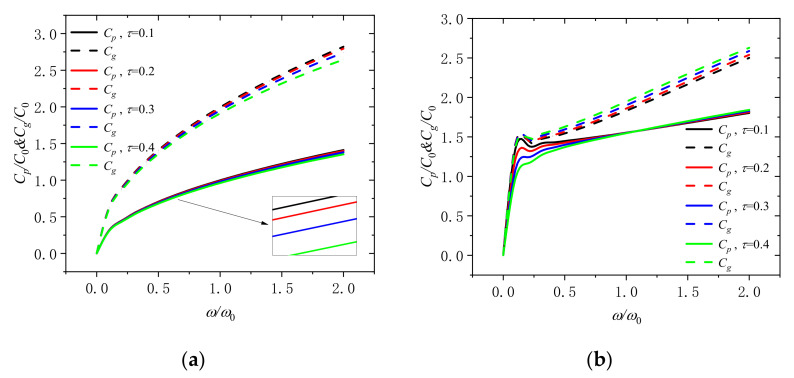
Phase speed and group speed in a rotating microbeam with different nonlocal scales (*τ* = 0.1, 0.2, 0.3, and 0.4) for (**a**) Ω/Ω_0_ = 0 and (**b**) Ω/Ω_0_ = 2.

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
