# Peer review of "Wave Propagation in Rotating Functionally Graded Microbeams Reinforced by Graphene Nanoplatelets"

_molecules, 2021, doi:10.3390/molecules26175150_

Round 1

Reviewer 1 Report

This is an interesting paper talking about numerical simulation on wave propagation in rotating functionally graded (FG) micro-beams reinforced by graphene nanoplatelets (GPLs). The graphene nanoplatelets (GPLs) are considered to distribute gradient in the diameter direction of the micro-beam, which leads to the functionally graded structure. The authors modelled rotating micro-beams and study the effects of different parameters. It can be published in Molecules after major revisions, my detailed comments are the following:

  1. It is better to clearly suggest in the introduction, what are those experimental studies and what are theoretical studies.
  2. There are some typos, for example RB=1?m, which cannot show the real unit of the parameter. Authors should clearly go through the manuscript to correct all typos.
  3. The authors should suggest the possible ways to create the GPL distribution patterns in experiment, so the simulation will be meaningful.
  4. How the material parameters are chosen, what is the background material without GPL reinforcement?
  5. In experiment, what is the possible way to control the LG/tG=10, 102, 103, 104?

Author Response

Dear Editors,

On behalf of my co-authors, we thank you very much for giving us an opportunity to revise our manuscript. We appreciate editors and reviewers very much for their positive and constructive comments and suggestions on our manuscript entitled “Wave propagation in rotating functionally graded micro-beams reinforced by graphene nanoplatelets”.

We have studied reviewer’s comments carefully and have tried our best to revise our manuscript according to the comments. The revision is marked in red in the manuscript. The detail point-to-point reply is listed at the bottom of this letter.

Once again, we would like to express our great appreciation to you and the reviewer for comments on our paper. Looking forward to hearing from you.

Thank you and best regards.

Yours sincerely,

Tian Yu Zhao

Northeastern University

Editor and Reviewer comments:

Reviewer 1

This is an interesting paper talking about numerical simulation on wave propagation in rotating functionally graded (FG) micro-beams reinforced by graphene nanoplatelets (GPLs). The graphene nanoplatelets (GPLs) are considered to distribute gradient in the diameter direction of the micro-beam, which leads to the functionally graded structure. The authors modelled rotating micro-beams and study the effects of different parameters. It can be published in Molecules after major revisions, my detailed comments are the following:

  1. It is better to clearly suggest in the introduction, what are those experimental studies and what are theoretical studies.

Reply: The paper investigates wave propagation in rotating functionally graded micro-beams reinforced by graphene nanoplatelets in theory. Thus, the references in the introduction are theoretical studies.

  1. There are some typos, for example RB=1?m, which cannot show the real unit of the parameter. Authors should clearly go through the manuscript to correct all typos.

Reply: The error has been revised (RB=1×10-6m).

  1. The authors should suggest the possible ways to create the GPL distribution patterns in experiment, so the simulation will be meaningful.

Reply: The preparation method for the GPL reinforced composite is given as follows.

The obtained materials are as follows.

(a) polymer matrix  (b) uniform GPL reinforced material  (c) FG-GPL reinforced material

  1. How the material parameters are chosen, what is the background material without GPL reinforcement?

Reply: Because rotating micro-beams are generally used in the micro aerial vehicle, the lightweight polymer material has great advantages. Moreover, GPLs are introduced to be as reinforcement to enhance the structural stiffness.

  1. In experiment, what is the possible way to control the LG/tG=10, 102, 103, 104?

Reply: Different materials preparation technologies can be adopted to obtained GPLs with various dimensions. Every size of GPLs is applied separately.

Reviewer 2

The introduction needs major revision. It should contain the significance of the problem under consideration, what has been done in the literature and what interesting results are found in previous works, what is the gap in our knowledge and what is the question the authors want to answer and what method they are going to use. In the current form, all is present is (authors did this... those authors did that) without explaining about their findings and what is missing in these previous works.

Reply: The introduction has been revised. (To sum up, most research are focused on the vibration performance of micro-beams. It is known that the microstructures are always failure due to the vibration. As an important method of nondestructive inspection, wave propagation characteristics in micro-beams is quite necessary to be carried out. To sum up, most research are focused on the vibration performance of micro-beams. It is known that the microstructures are always failure due to the vibration. As an important method of nondestructive inspection, wave propagation characteristics in micro-beams is quite necessary to be carried out. For enhance the structural performance, composite materials are always adopted in a real project.)

The authors seem to use a template for this journal and forgot to remove the guidelines from their text. For example, in section 2 lines 69 and 70 are irrelevant to the rest of the text and should be removed.

Reply:

Overall, the write-up of the paper looks more like a class project report, rather than a scientific paper and requires extensive polishing to be appropriate for publication.

Reply: The paper has been double checked.

Reviewer 2 Report

The introduction needs major revision.  It should contain the significance of the problem under consideration, what has been done in the literature and what interesting results are found in previous works, what is the gap in our knowledge and what is the question the authors want to answer and what method they are going to use. In the current form, all is present is (authors did this... those authors did that) without explaining about their findings and what is missing in these previous works.

The authors seem to use a template for this journal and forgot to remove the guidelines from their text. For example in section 2 lines 69 and 70 are irrelevant to the rest of the text and should be removed.

Overall, the write-up of the paper looks more like a class project report, rather than a scientific paper and requires extensive polishing to be appropriate for publication. 

Author Response

(The authors gave the same response as above.)

Round 2

Reviewer 2 Report

I only have one comment.

This paragraph in the introduction is repeated: "

To sum up, most research are focused on the vibration performance of micro-beams.  It is known that the microstructures are always failure due to the vibration. As an important method of nondestructive inspection, wave propagation characteristics in micro beams is quite necessary to be carried out.

"

Author Response

Dear Editors and reviewers,

On behalf of my co-authors, we thank you very much for giving us an opportunity to revise our manuscript. The repeated paragraph in the introduction has been deleted.

Kinds,

Tianyu Zhao